# Are Integrins Still Practicable Targets for Anti-Cancer Therapy?

**DOI:** 10.3390/cancers11070978

**Published:** 2019-07-12

**Authors:** Begoña Alday-Parejo, Roger Stupp, Curzio Rüegg

**Affiliations:** 1Pathology Unit, Department of Oncology, Microbiology and Immunology, Faculty of Science and Medicine, University of Fribourg, Chemin du Musée 18, Per 17, CH-1700 Fribourg, Switzerland; 2Departments of Neurological Surgery, Malnati Brain Tumor Institute of the Lurie Comprehensive Cancer Center, Neurology and Medicine, Feinberg School of Medicine, Northwestern University, 676 N St Clair St, Suite 2210, Chicago, IL 60611, USA

**Keywords:** cancer, angiogenesis, tumor microenvironment, imaging, therapy

## Abstract

Correlative clinical evidence and experimental observations indicate that integrin adhesion receptors, in particular those of the αV family, are relevant to cancer cell features, including proliferation, survival, migration, invasion, and metastasis. In addition, integrins promote events in the tumor microenvironment that are critical for tumor progression and metastasis, including tumor angiogenesis, matrix remodeling, and the recruitment of immune and inflammatory cells. In spite of compelling preclinical results demonstrating that the inhibition of integrin αVβ3/αVβ5 and α5β1 has therapeutic potential, clinical trials with integrin inhibitors targeting those integrins have repeatedly failed to demonstrate therapeutic benefits in cancer patients. Here, we review emerging integrin functions and their proposed contribution to tumor progression, discuss preclinical evidence of therapeutic significance, revisit clinical trial results, and consider alternative approaches for their therapeutic targeting in oncology, including targeting integrins in the other cells of the tumor microenvironment, e.g., cancer-associated fibroblasts and immune/inflammatory cells. We conclude that integrins remain a valid target for cancer therapy; however, agents with better pharmacological properties, alternative models for their preclinical evaluation, and innovative combination strategies for clinical testing (e.g., together with immuno-oncology agents) are needed.

## 1. Introduction

Since their initial discovery as extracellular matrix (ECM) cell adhesion receptors over 30 years ago, integrins were rapidly identified as molecules relevant to cancer cell functions, notably migration, invasion, and metastasis formation. Cancer and leukocyte biology research greatly contributed to unraveling many of the cellular and molecular features of integrins as we know them today [1,2]. The characterization of their molecular structure, activation, and signaling functions, lead to fundamental discoveries with far-reaching implications in many fields of biology and medicine [3,4]. The development of integrin inhibitors based on the Arg–Gly–Asp binding sequence, raised great hopes for the development of novel anti-cancer therapies, in particular to inhibit tumor angiogenesis [5]. Despite encouraging results in preclinical models, all efforts to translate the experimental results into a therapeutic benefit for cancer patients were disappointing, and can be illustrated by the failure of the αVβ3/αVβ5 inhibitor cilengitide as an anti-cancer drug [6,7]. This integrin inhibitor has gone through a full preclinical and clinical development cycle, and ultimately failed in randomized trials in several disease entities. At this point, a fundamental question is warranted: are integrins still practicable therapeutic targets in cancer, despite the failure of targeting αVβ3/αVβ5 and α5β1 in several clinical trials? We need to re-evaluate the role of integrins in cancer, including how can we best target them, and how we can translate preclinical observations into clinical benefits. Here, we will review selected aspects of integrin biology and cancer-related function, and discuss some considerations for future developments as anti-cancer therapeutics aiming at lessons learned.

## 2. Integrin Adhesion Receptors, a Class of Its Own

Integrins are αβ heterodimeric cell surface adhesion receptors. There are 18 α and eight β subunits consisting each of a long extracellular domain (750–1000 amino acids), and short transmembrane and cytoplasmic domains (20–75 amino acids, except for the β4 cytoplasmic subunit up to over 1000 amino acids long), which in combination form 24 different heterodimers [8,9]. Integrins bind to insoluble ECM proteins (e.g., fibronectins, laminins, collagens), matricellular proteins (e.g., Cyr61/CTGF/NOV, CCN), cell surface (e.g., Intercellular Adhesion Molecules, ICAMs; Vascular Cell Adhesion Molecule-1, VCAM-1) and soluble (e.g., fibrinogen, complement proteins, Vascular Endothelial Growth Factor, VEGF; Fibroblast Growth Factor 2, FGF2; angipoietin-1 or Transforming Growth Factor β, TGFβ) [9,10] ligands. Binding occurs through a pocket formed by the α and β subunits or through the I-domain on some α chains [11]. The ligand binding specificity is promiscuous (one integrin binds multiple ligands) and redundant (different integrins bind to the same ligand) [12]. Promiscuity may be advantageous in conditions when function is more important than the specificity of the eliciting event. This is the case during wound healing, where cells have to cope with a rapidly changing ECM. Integrin αVβ3, which binds nearly a dozen of different ligands, is the prototype of a promiscuous integrin. Redundancy may reflect the need for a given cell to respond differently to the same ECM protein. For instance, α5β1 and αVβ6 bind to fibronectin, but elicit different responses [13]. Integrins exist in a low affinity, closed (bent) form and a high affinity, active, open (extended) form. Integrin activation involves the binding of two cytoplasmic adaptor proteins, talin and kindlin, to the intracellular domain of the β-integrin (“inside-out signaling”). In turn, high-affinity ligand binding induces a further conformational change of the cytoplasmic tails, promoting linkage to the actin cytoskeleton, focal complexes formation, and signaling events that are required for stable cell adhesion, spreading, migration, proliferation, survival, and differentiation [11,14]. Many integrins expressed on cancer cells or cells of the tumor microenvironment have been reported to be involved in cancer progression. An overview is given in Table 1.

## 3. Integrin Adhesome as a Signaling Complex

Integrins lack intrinsic enzymatic activity and rely on structural (e.g., paxillin, vinculin) and adaptor (e.g., SRC Homologous and Collagen-like (SHC), Crk-Associated Substrate, p130 (CAS)) proteins and enzymes (e.g., Focal Adhesion Kinase, FAK; Integrin Linked Kinase, ILK) for their signaling function. Many of the signaling pathways activated by integrins are also activated by growth factor receptors (GFRs), such as epidermal growth factor receptor (EGFR) or vascular growth factor receptor (VEGFR). Maximal signaling is achieved when GFRs and integrins are engaged. The molecular basis of this cooperation is believed to be the ability of engaged integrins to cluster intracellular adaptor and signaling proteins, thereby facilitating interactions with GFRs [33,34]. In addition, integrins can physically associate laterally with cell membrane proteins (e.g., CD151 or CD47) to elicit or modulate signaling events [35]. This complex and highly dynamic structure at the interface between cell adhesion and GFRs signaling is referred to as the adhesome [36]. Signaling integration provides enhanced specificity, as well as temporal and spatial control over many cellular events, compared to signaling from GFRs alone. Integrins activate four main signaling pathways relevant to cancer initiation, progression, metastasis, angiogenesis, and inflammation: the Rat Sarcoma (RAS)-mitogen activated protein kinases (MAPKs), the phosphoinositid-3-kinase (PI3K)-AKT, the Rho-family GTPases, and the Nuclear Factor kappa B (NF-κB) pathways (for more insights and details, we refer to recently published reviews [37,38,39,40,41,42].

## 4. Cancer Cell Integrins

Soon after their discovery, it was observed that integrin expression is altered in cancer compared to the corresponding healthy tissue, particularly of α3β1, α4β1, α5β1, α6β4, αvβ3, αvβ5, αvβ6, and αvβ8 [28], and that in certain cancers, this altered expression was correlated with outcomes [43]. The complexity of integrin regulation, ECM composition, concomitant signaling from GFRs, and pleiotropic functions, together with the possibility that their contribution to cancer may differ across different cancer types, stages, and treatment, significantly complicated assigning a specific cancer-related phenotype to a given integrin. Observations about their clinical relevance were merely correlative and often contradictory [16,18,19,44]. Nevertheless, some key contributions to cancer progression, in particular metastasis, have been established. For a general overview of cancer-related functions, in particular to the metastatic cascade, we refer to some recent comprehensive reviews [13,20,23,28,30,37,45]. Here, we will discuss selected known and emerging roles of integrins and their relevance to events critical for cancer progression (Figure 1).

### 4.1. Epithelial-to-Mesenchymal Transition (EMT) and Cancer Invasion

Epithelial cancer cells can undergo a complex, multistep gene expression reprogramming process referred to as epithelial–mesenchymal transition (EMT), culminating with the down-regulation of epithelial-specific genes and the up-regulation of mesenchymal specific ones, which are associated with increased motility, invasion, and metastasis [46]. Integrins play an important role in the induction of EMT and in mediating some of its effects. For instance, integrin α3β1 expression is required for TGFβ1-stimulated Small Mothers Against Decapentaplegic (SMAD) signaling, leading to EMT [47,48]. On the other side, epithelial cells stimulation with TGFβ1 leads to a down-regulation of β4 integrin, which is a typically epithelial integrin essential for epithelial integrity and stability, resulting in facilitated migration (see Figure 1) [49]. The signaling axis PEAK1/ZEB1 is mediating TGFβ1-induced EMT in breast cancer through integrin β3 and fibronectin interaction [50]. In turn, integrins are critical mediators of EMT. For example, the transcription factors TWIST1 and AP-1 cooperate to upregulate integrin α5 expression to induce EMT and tissue invasion [51]. Mechanisms of invasion are quite universal and shared by most tissues and cells. They include guidance by integrins toward fibrillar collagen and/or laminins, haptotactic migration though chemokines and growth factors, and physical pushing [52]. Cancer cell invasion occurs preferentially along pre-existing ECM tracks of least resistance, followed by tissue remodeling. For example, invasive breast cancer cells preferentially invade along collagen bundles and adipocytes. Matrix remodeling/degradation, in concert with integrins function, is key to invasion. The elongation factor-2 kinase (eEF-2K) regulates the invasive phenotype of pancreatic cancer cells by activating a signaling axis consisting of tissue transglutaminase (TG2) and the β1 integrin/uPAR/MMP-2 complex as well as a decrease in SRC activity [53]. The αVβ3 integrin in turn controls matrix metalloproteases 9 (MMP9) activity during invasion by binding to its Hemopexin (PEX) domain, resulting in controlled pericellular proteolysis [54]. Invasive migration and proteolytic remodeling of the ECM are interdependent processes that control tissue micropatterning events that are critical for the transition from collective to individual cancer cell invasion [55]. Importantly, cancer cells do not only invade through integrin-mediated interaction with the ECM, but can also do so by ameboid migration by a push-and-pull mechanism in the absence or integrin engagement and metalloproteases [35,56,57].

### 4.2. Anoikis

Anchorage-dependent cell survival and growth are essential functions of integrin-mediated adhesion to the ECM. Upon cell detachment from the ECM, integrin clustering and the adhesome are disrupted, resulting in the dispersion of GFRs, loss of cell signaling, and cell death (anoikis) [58]. However, recent studies suggest that in non-adherent cells, integrins may still be in an active signaling state through the binding of ECM fragments or soluble ligands, and contribute to promote anchorage-independent cancer cell survival, growth, and metastasis. Active FAK, which is a critical transducer of integrin-mediated survival, is observed in β1 integrin-positive endosomes, where it may contribute to initiate cell survival signals in cells in suspension [59]. Similarly, αVβ3-positive adhesomes promote SRC-CAS (Rous sarcoma oncogene cellular homolog-Crk-associated substrate)-dependent cell survival independently of FAK and cell adhesion, thereby pointing to a role for αVβ3 anchorage-independent tumor cell growth, survival, and aggressiveness [60]. CCN proteins associate to the ECM and bind to integrins or cell surface proteoglycans to regulate cell proliferation, motility, and differentiation [10]. We have previously shown that the matricellular CCN1 (CYR61) promotes the lung metastasis of triple negative breast cancer (TNBC) cells by binding to active β1 integrin at the cell surface and promoting AMP-activated protein kinase α (AMPKα) signaling, survival, and early colonization of the lung [61]. Recent observations indicate that integrin–GFRs cross-talk may persist in detached cells to provide survival signals. cMET (or Hepatocyte Growth Factor Receptor) activation stimulates the endocytosis of active β1 integrin, which in turn sustains cMET signaling to the MAPK pathway, resulting in anchorage-independent growth and metastasis (see Figure 1) [62]. This observation, in addition to the previously reported cross-talk between the β1 and β4 integrins and cMET signaling, suggests that targeting integrins may enhance the anti-tumor activity of cMET inhibition in adherent and non-adherent cancer cells [63,64]. Thus, the role of integrins in promoting survival may not be restricted to conditions of cell adhesion to the ECM, but may further continue once cells are detached, thereby opening unanticipated therapeutic options.

### 4.3. Metabolism

Several pathways controlling metabolic functions, such as such as AMPK, mammalian target of rapamycin (mTOR), and hypoxia-inducible factor (HIF) in response to nutrients’ availability and needs, also control integrin expression and function [65,66]. In turn, integrins control metabolic functions, thereby establishing a reciprocal, dynamic communication between cell adhesion and metabolism [67]. Metabolic events regulate integrin expression and function at the levels of transcription, degradation, recirculation, and glycosylation. These mechanisms and their impact on integrin biology have been recently summarized and discussed in excellent review articles, and thus will not be addressed here [67,68]. Instead, we will briefly highlight the modulation of metabolism by integrins. β1 integrins activate the PI3K–AKT pathway via FAK or ILK, AKT, and mammalian Target of Rapamycin Complex 1 (mTORC1) [42]. mTORC1 is a critical regulator of the reprogramming of lipids, nucleotides, and amino acids metabolisms, and an inducer of the EMT promoter TWIST [69,70,71]. Integrin-mediated adhesion, PI3K–AKT signaling, and MAPK signaling promote cell survival during times of nutrient deprivation, for instance in hypoxic tumor regions, through the induction of autophagy [72]. Integrin–ILK dependent signaling also controls the Hippo signaling pathway, which is a critical nutrient-sensing system, leading to the Yes-Associated Protein/Tafazzin (YAP/TAZ)-dependent transcription of proliferation and cell survival genes. It is interesting that tenascin-C (TNC), an ECM protein promoting invasion and metastasis, modulates YAP via binding to α9β1 integrin [73]. Thus, integrin signaling during EMT contributes to coordinate the concomitant global changes in cell metabolism that are relevant for cell proliferation, migration, and survival [74,75,76,77]. Integrin signaling is also involved in the metabolic reprogramming of cancer cells from oxidative phosphorylation toward glycolysis and biosynthesis. TWIST induces aerobic glycolysis in breast cancer cells via β1 integrin signaling through the FAK–PI3K–Akt–mTOR axis [78]. Consistent with this, integrins also control glucose transport, and in breast cancer cells, the loss of adhesion reduces glucose uptake, ATP production, and fatty acid oxidation [79]. In migrating cells, β1 integrin interacts with the lactate transporter Mono Carboxylate Transporter 4 (MCT4) at the leading edge and integrin-mediated cell migration depends on MCT4 function to export excessive lactic acid and control intracellular pH (see Figure 1) [80,81]. Thus, there is emerging evidence indicating that integrins are involved in the regulation of metabolic functions at steps when cancer cells alter their interaction with the ECM or acquire novel integrin-dependent activities (e.g., motility, invasion, survival).

### 4.4. Stemness and Resistance to Therapy

Similar to normal tissues, cancers are hierarchically organized and contain cells with stem cell-like features, which are referred to as cancer stem cells (CSCs), which can drive tumor initiation, self-renewal (maintenance), resistance to therapy, relapses, and metastasis [82,83]. They are present in specific locations (niches) that are rich in particular ECM proteins such as periostin and TNC, and close to vascular cells [84,85]. By interacting with the surrounding ECM, integrins appear to control the balance between physiological stem cell renewal and differentiation [86]. It is of interest that most integrins that have been associated with physiological stem cells are also expressed in CSCs [22,83]. In particular, α6 integrins (i.e., α6β1 and α6β4) are widely expressed in CSCs in breast [87], prostate [88], colorectal [89], brain [90], and non-small cell lung [91] cancers. While integrin expression patterns have been initially used as markers to identify CSCs, it is increasingly clear that they play functional roles. Integrin α6 contributes to breast cancer initiation by inducing FAK-mediated expression of the polycomb complex protein B cell-specific Moloney murine leukemia virus Integration site 1 (BMI-1), which is necessary for CSCs self-renewal, via its cytoplasmic domain [92,93]. Integrin β3 is essential for maintaining the CSC phenotype in breast [94], pancreas, and lung [95] cancers. αVβ3 expression distinguishes mammary luminal progenitors (β3^high^) from mature luminal cells (β3^low^) [96], and αVβ3 contributes to mediating CSC properties, including spheroid formation, tumor initiation [97], and metastasis [60]. Targeting CSCs integrins to modulate their interaction with the ECM of the stem cell niche may be of therapeutic potential. For instance, the inhibition of integrin α6 suppressed the CSC phenotype and impacted cancer progression in glioblastoma [90]. Thus, the modulation of CSCs function and fate is a novel emerging function of integrins in cancer (see Figure 1).

The role of integrins in CSC biology is likely to contribute to chemoresistance and tumor relapse, which is an integrin effect that was reported earlier [98]. In breast cancer, paclitaxel (Taxol) treatment enriches high CSC for integrin α6 [93], and in a spontaneous lung cancer model, CSCs expressing integrin β4 are enriched after cisplatin treatment [91]. Integrin β3 is highly expressed in cancer cells with acquired resistance to the EGFR inhibitors to erlotinib and lapatinib through the activation of NF-κB signaling [95]. Integrin β1 has been reported to promote resistance to radiotherapy in head and neck cancer [99], lapatinib and trastuzumab resistance in breast cancer [100], and erlotinib resistance in lung cancer [101] by enhancing SRC and AKT activities. Consistently, silencing integrin β1 restored erlotinib sensitivity [101]. DNA damage can enhance αVβ3 expression on resistant cells, facilitating clearance by phagocytosis, and thereby dumping the immune response [102]. Consistently, the inhibition of αVβ3-mediated phagocytosis enhanced antibody-dependent cytotoxic responses [103]. Thus, unexpectedly, tumor cell αVβ3 may turn out to be a regulator of the anti-tumor immune response, thereby opening new therapeutic opportunities.

### 4.5. Metastatic Niche

Tumor cells leaving the primary tumor on the way to form metastases face their main challenge when they enter the distant organ, and need to adapt to survive in the new tissue [104]. For this, disseminated tumor cells (DTC) rely on a specialized microenvironment called the metastatic niche, promoting their survival and outgrowth [105,106]. The recruitment of inflammatory/bone marrow-derived cells and endothelial cells, through the production of growth factors, cytokines, and chemokines, the modification of the ECM, and, paradoxically, hypoxia, are essential elements of the niche [107,108,109,110]. Metastatic niches can be induced by primary tumors even before DTC reach the peripheral tissues, and are therefore also referred to as pre-metastatic niches. This implies a cross-talk between the primary tumor and peripheral tissues [111,112]. Inflammation is a key element of this cross-talk and the (pre)metastatic niche formation. Pro-inflammatory factors, such as S100 or Serum Amyloid A acute phase Proteins (SAP) family members induced by the primary tumor, play a critical role in the formation of (pre)metastatic niches, including the recruitment of CD11b+ myeloid cells [113]. Exosomes released by cancer cells promote metastasis by contributing to the formation of organ-specific (pre)-metastatic niches through their ability to transfer metabolites, proteins, and RNA to distant tissues [114,115,116]. Strikingly, integrins enriched on the surface of cancer-derived exosomes contribute to organ-specific targeting. For instance, exosomes shed by cancer cells metastasizing to the lung are enriched for α6β1 and α6β4 integrins, which home themselves to the lung, while αVβ5 integrin-rich exosomes shed by liver-tropic cancer cells are that preferentially home to the liver (see Figure 1) [116]. Once in the target organ, exosomes actively contribute to the formation of the (pre)metastatic niche by inducing the expression of specific ECM proteins and pro-inflammatory factors, including S100 proteins favoring the recruitment of inflammatory cells [116]. Further research is necessary in order to evaluate the clinical significance of these observations, and in particular whether impinging on the integrins-mediated homing of tumor-derived exosomes may have an adjuvant therapeutic effect on metastatic tumor progression. On the other side, circulating exosomes may be exploited to identify patients progressing to metastasis.

### 4.6. Metastatic Dormancy

DTC can remain quiescent for prolonged periods of time as single cells, small clusters, or micrometastases before resuming growth to form macrometastases in a state called metastatic dormancy [117,118]. This is particularly relevant for breast cancer, where clinical relapses can occur years or decades after primary cancer therapy [119,120]. Interaction with the ECM is implicated in controlling dormancy, paralleling the role of cell–ECM interaction in physiological CSC niches [121]. For example, high Urokinase-type Plasminogen Activator Receptor (uPAR) expression and α5β1 integrin binding to fibronectin suppresses p38 activity, increases ERK activity, and promotes cell proliferation. Accordingly, low uPAR-expressing cells have a high p38/Extracellular Regulated Kinase (ERK) activity ratio, fail to assemble fibronectin fibrils and ligate α5β1 integrin, and are dormant in vivo [122]. Similarly, the loss of α5β1 integrin expression results in the inactivation of the RAS–Rat Fibrosarcoma (RAF)–ERK signaling pathway, the activation of p38/Janus N-Terminal Kinase (JNK) stress signaling pathway, the induction of the TP53/RB-dependent cell-cycle arrest, and dormancy (see Figure 1) [123]. This suggests a role for the cross-talk between mitogenic and stress signals regulated by the uPAR–α5β1–ECM axis in controlling cellular dormancy [124]. Interestingly, collagen-rich (fibrotic) ECM promotes the transition of dormant DTC to growing DTCs [121,125,126,127]. Accordingly, β1 integrin ligation mediates the awakening of dormant DTC in a murine breast cancer model [121]. The ECM protein periostin, which is a αVβ3 and αVβ5 ligand present in the primary and metastatic tumor stroma, drives DTC escape from dormancy by activating Wingless Int-1 (WNT) signaling [128,129]. Inhibition of the PI3K–AKT signaling cascade, which is a pathway also controlled by integrins, can activate autophagy and induce quiescence [130], while low or absent AKT signaling in DTC correlates with dormancy in breast cancer patients [130,131]. Consistently, dormant tumor cells express high levels of Aplysia Ras Homology Member I (ARHI), which is an inhibitor of the PI3K–AKT cascade, and ARHI silencing breaks dormancy in several experimental models [132,133]. Thus, controlling cancer dormancy is emerging as an unanticipated activity of integrins, and interfering with ECM integrins interaction may be a therapeutic approach to consider in order to promote cancer dormancy [134,135]. Further studies are warranted to unravel their potential operability in patients at risk for progression or recurrence after initial therapy, particularly radiotherapy.

## 5. Tumor Stroma

The tumor microenvironment (TME) contains a multitude of cells that positively or negatively impact tumorigenesis, tumor growth, invasion, and metastasis, two of which are fibroblasts and endothelial cells [136]. Integrins expressed on these cells participate in the cross-talk relevant to tumor progression.

### 5.1. Fibroblasts and the Extracellular Matrix

Altered composition of the tumor ECM, such as increased fibrillar collagen deposition, cross-linking, and rigidity provide guidance cues and oncogenic signals for cancer cell growth and invasion in multiple cancers, including breast, colorectal, head and neck, and pancreas [137]. Matrix cross-linking through lysil oxydases (LOXs) increases matrix stiffness, integrin-dependent signaling, and SRC-dependent cell proliferation, resulting in facilitated tumor progression and metastasis [30,138,139,140]. Altered collagen deposition, ECM modification, and increased cancer-associated fibroblasts (CAF) contractility may be a general hallmark of tumor progression and poor prognosis, and therefore a potential therapeutic target [137]. We recently reported a novel mechanism by which CAF induces contact-dependent Colorectal Cancer CRC cell motility and invasion. Activated fibroblasts express FGF-2 on their surface and present it to FGF receptors (FGFR) on CRC cells, resulting in integrin αVβ5-dependent CRC cell migration along fibroblasts. The inhibition of FGF-2 on fibroblasts or FGFR, SRC, and αVβ5 on cancer cells prevented these effects (see Figure 1) [141]. By using an orthotopic model of CRC, we validated in vivo these in vitro results. The co-injection of CRC cells with fibroblasts in the cecum of mice promoted lung metastasis, which was prevented by treatment with the SRC and FGFR kinase inhibitors dasatinib and erdafitinib, respectively [142]. These experiments suggest that the FGF2–FGFR–SRC–αVβ5 integrin axis might be a potential therapeutic target to prevent metastasis in stage II and III CRC. In a previous study, we identified the matricellular protein CCN1 and αVβ5 integrin as proteins cooperating to mediate the invasion and metastasis of tumors growing in hypoxic pre-irradiated tissues. αVβ5 inhibition by a pan-anti-anti-αV monoclonal antibody (mAb) 17E6 or the αVβ3/αVβ5-specific cyclic Arg–Gly–Asp peptide cilengitide [143] attenuated CYR61/CTGF/NOV1 (CCN) 1-dependent metastasis [144]. Thus, integrins are important mediators of the interaction of tumor cells, CAF, and matricellular proteins relevant to tumor progression, and are therefore of potential therapeutic relevance.

### 5.2. Endothelial Cell

αVβ3 was the first integrin reported to be preferentially expressed in angiogenic endothelial cells [145]. The inhibition of αVβ3 through antibodies, Arg–Gly–Asp-based cyclic peptides, or non-peptidic mimetics suppressed tumor angiogenesis without affecting quiescent endothelial cells. In preclinical studies, the inhibition of angiogenesis with αVβ3 antagonists suppressed tumor progression, raising a high expectation that αVβ3 inhibition may be a valuable anti-cancer strategy (see Figure 1) [9,146]. However, genetic ablation of the αV or β3 subunit had minimal impact on developmental angiogenesis [147,148], while it increased VEGFR-2 signaling and tumor angiogenesis [149,150]. Interestingly, the acute genetic deletion of endothelial cell αVβ3 transiently suppressed tumor angiogenesis if performed before tumor implantation, but not once tumors were already growing [151]. Low concentrations of high affinity Arg–Gly–Asp-based peptidic inhibitors such as cilengitide induce αVβ3 affinity maturation and signaling, resulting in stimulated angiogenesis [152]. These observations question the relevance of αVβ3 as a target in anti-angiogenesis therapies. High-affinity inhibitors disrupt the Vascular Endothelial (VE)–cadherin junction, and increase permeability through αVβ3 activation and FAK–SRC signaling in vitro [153]. These effects translated into increased vascular permeability of αVβ3-positive tumor vessels in tumor-bearing mice treated with cilengitide, resulting in increased chemotherapy delivery to the tumor relative to healthy tissue [154]. Interesting, the αVβ3 function appears to be modulated by inflammatory factors. A combined administration of high doses of Tumor Necrosis Factor/Interferon gamma (TNF/IFNγ) through an isolation limb perfusion setting to cancer patients with sarcomas or melanoma metastases of the limbs inactivates endothelial cell αVβ3, causing endothelial cell death and selective disruption of the tumor vasculature [155]. Inactive αVβ3 integrin acts permissive for TNF to kill endothelial cells through a lack of AKT activation and anti-apoptotic signals [156]. Prostaglandin E2 (PGE_2_) promotes tumor angiogenesis by activating αVβ3 function and signaling through the prostane receptors cyclic Adenosine Monophosphate (cAMP), Protein Kinase A (PKA) and RAC, and these effects are blocked by Cyclooxygenase-2 COX-2 inhibition [157]. However, endothelial cells express many additional integrins beyond αVβ3/αVβ5, including α4β1, α5β1 (fibronectin receptors), α9β1 (tenascin receptor), α3β1, α6β1 and α6β4 (laminin receptors), and α1β1 and α2β1 (collagen receptors) [158]. Due to the overlap of integrin expression across different cell types and tissues, the selective targeting of integrins to inhibit tumor angiogenesis remains challenging. Nevertheless, the inhibition of α1β1, α2β1, α5β1 suppressed tumor angiogenesis and reduced tumor growth in many experimental models [9,159].

However, clinical trials with the αVβ3 integrin inhibitors cilengitide [143] failed to demonstrate significant therapeutic benefits, including in highly angiogenic glioblastoma [160]. Considering the failures of other anti-angiogenic therapies (e.g., the anti-VEGF antibody bevacizumab) in halting glioblastoma progression, the failure of cilengitide as anti-angiogenic drug [143] may not be due to cilengitide itself, but rather to limitations intrinsic to all anti-angiogenic approaches, in particular evasive resistance [161,162,163].

In summary, integrin inhibition as an anti-cancer therapy was initially conceived based on the role of integrins in promoting cancer cell invasion, metastasis, and tumor angiogenesis. Recent developments indicate that additional functions relevant to cancer cells are also mediated or regulated by integrins, including dormancy, metabolism, survival, therapy resistance, EMT, fibrosis, cancer cell stemness, exosome homing, and pre-metastatic niche formation. Thus, in the future, it will be important to understand the contribution of integrins to these emerging functions, and evaluate the potential therapeutic impact of impinging on these functions by inhibiting integrins.

## 6. Targeting Integrins in Cancer

Antibodies, endogenous proteins, peptidic antagonists, synthetic peptides, and peptidomimetics have been used to target integrins in cancer. Some of these molecules were or are still in clinical development, but none of them have been successfully established as an anti-cancer agent to date [30,164].

### 6.1. Inhibiting Integrin Function

Anti-αVβ3 antibody etaracizumab (MEDI-522) entered phase I and II clinical studies and showed good tolerability, also in combination with chemotherapy, but no anti-angiogenic or immunomodulatory effects were noted [165,166]. Anti-αV antibodies, such as intetumumab (CNTO95) or abituzumab (EMD 525797/DI17E6) entered phase I and II clinical testing as single agents or in combination with cytotoxic agents and/or other targeted molecules. Clinical trials were initiated in a variety of solid tumors, including melanoma, sarcoma, colorectal, and prostate cancers [167,168]. Abituzumab showed specific activity in prostate cancer bone metastases [169]. A randomized, double-blinded phase 2 study of abituzumab in combination with the EGFR inhibitor cetuximab and Laucovorin, Fluorouracil, Irinotecan (FOLFIRI) chemotherapy in first-line RAS^WT^ metastatic CRC with high αVβ6 integrin expression is planned and expected to be completed in August 2021 (www.clinicaltrials.gov/NCT03688230). The anti-α5β1 integrin antibody M200/volociximab was shown to inhibit angiogenesis and suppress tumor growth and metastasis in mice [170]. It entered clinical testing as a single agent in advanced epithelial ovarian or primary peritoneal cancer [171] and in combination with chemotherapy in advanced non-small-cell lung cancer (NSCLC) [172]. Volociximab was generally well tolerated, and showed preliminary evidence of efficacy in advanced NSCLC. Endogenous antagonists such as the peptides endostatin, tumstatin, or angiostatin showed anti-cancer activity profiles [9]. Recombinant endostatin, a supposed α5β1 inhibitor, was tested in clinical studies in combination with chemotherapies and radiotherapies [173,174], but the results have been inconsistent, which also relates to problems in the production of the active protein [175].

The Arg–Gly–Asp-based cyclic peptide cilengitide (EMD121974) targeting αVβ3/αVβ5 has been the most advanced and investigated integrin inhibitor so far [143]. In spite of preclinical evidence of anti-cancer activity and great expectations from phase II clinical studies [176], phase III studies, in combination with chemotherapy, targeted agents, or radiotherapy in a multitude of cancers—most notably glioblastoma—failed to provided clinical benefits [6,7,160,177,178]. The α5β1-blocking non Arg–Gly–Asp-based peptide ATN-161 entered clinical testing [179] based on preclinical anti-cancer activities [180,181], but also failed to provide therapeutic benefits. It is currently being investigated in combination with VEGF inhibition for the treatment of wet age-related macular degeneration [182].

Peptidomimetics are synthetic compounds mimicking the structure and action of natural peptides that have the advantages of being insensitive to protease degradation, able to be administered orally, and having longer stability. Many peptidomimetics targeting αVβ3, αVβ5, and α5β1, including SCH221153, BCH-15046, SJ749, and JSM6427 have been developed and showed anti-cancer activities in preclinical models [9,164].

### 6.2. Targeting Drug to the Tumor

A conjugation of integrin-targeting antibodies, Arg–Gly–Asp-based cyclic peptides, or peptidomimetic, has been explored to improve the delivery and tumor uptake of drugs, biologicals, nanoparticles, and liposomes compared to unconjugated drugs [183,184]. For example, αVβ3-specific Arg–Gly–Asp-based cyclic peptides targeting the tumor vasculature or tumor cells have been successfully used to deliver therapeutic compounds, as well as image tumor lesions (theranostics) [185,186]. In general, these approaches have demonstrated superior ability in increasing drug uptake and activities in the tumors (reviewed in [185,187,188,189,190]). Dual targeting has further significantly improved drug delivery and activity. Utilizing peptides against P-selectin and αVβ3, which are two molecules that are functionally implicated in different stages of the metastatic disease, significantly increased drug delivery at metastatic sites, compared to a single molecule targeting [191]. Similarly, dual αVβ3 (vascular)/CD44 (cancer cell) targeting resulted in enhanced targeting efficiency and anti-tumor activities through the enhanced permeation and retention effect [192,193]. Cell-penetrating peptides can be combined with targeting peptides to improve drug delivery. For instance, tumor targeting through a tumor-specific peptide, followed by proteolytic cleavage and binding to a second receptor, improved delivery by facilitated extravasation and transport through extravascular tumor tissue [194]. Magnetic nanoparticles (magnetosomes) coupled to Arg–Gly–Asp peptides were targeted to tumors after systemic administration and used to generate therapeutic heat upon laser excitation, which successfully inhibited tumor progression [195].

### 6.3. Tumor Imaging

Integrin αVβ3, α5β1, and αVβ6 have been explored for non-invasive tumor imaging purposes, using magnetic resonance imaging (MRI), positron emission tomography (PET), computer tomography (CT), and optical and ultrasound-based imaging techniques [9,196,197]. Their targeting is of great potential relevance for the early diagnosis, staging of disease, patient’s stratification, and therapeutic monitoring [198]. The imaging of experimental tumors using modified αVβ3-binding Arg-Gly-Asp (RGD)-based peptides is a standard benchmark for the validation of tumor-targeting approaches [28]. PET technology has been preferred for use in animal models and in humans because of its high intrinsic sensitivity. The level of expression of αVβ3 detected by PET correlated with the level of αVβ3 determined by immunohistochemistry, suggesting that this approach is a good surrogate of integrin expression in vivo [197,199,200]. Tracers targeted by anti-αV integrin antibodies or peptides showed high levels of specific tumor accumulation by PET, Single Photon Emission Computed Tomography (SPECT), or optical imaging in different cancer models [201,202,203,204,205]. In spite of these interesting results, the clinical value of the in vivo mapping of αVβ3 to quantify tumor angiogenesis has not been clinically validated yet, as αVβ3 expression itself does not necessarily correspond to angiogenic activity in tumor tissues [200,206]. Integrin α5β1 has also been explored for PET and SPECT-based imaging of experimental tumors using peptidomimetic, linear, or cyclic peptides [207,208,209], while α5β1-based imaging in human has not been reported yet [28]. αVβ6-based imaging may be attractive, as it is associated with the invasion and activation of the TGFβ pathway in tumor cells. Using a αVβ6 peptide ligand identified by phage display library screening PET/CT-based scans successfully imaged head and neck cancers and NSCLC [210]. Interestingly, Nieberler et al. used a highly potent αVβ6-selective integrin ligand [211] for fluorescence-assisted intraoperative assessment of resection margins in patients with bone-infiltrating squamous cell carcinoma of the head and neck [212]. This approach could become an invaluable intraoperative guidance tool for the surgeons to assure tumor-free resection margins.

## 7. Open Questions and Challenges Ahead

Over 30 years of experimental research has provided compelling evidence that integrins are important mediators of cancer progression, and preclinical results indicate that they are potentially valuable therapeutic targets, in particular αVβ3 and α5β1, for anti-cancer therapies. Yet, to date, numerous clinical studies have failed to translate preclinical expectations into therapeutic benefits for patients (Table 2). The failure of cilengitide, the integrin antagonist that has been most widely tested in randomized clinical studies [143], was a major deception. After a large phase III trial evaluating cilengitide in combination with radiation and temozolomide chemotherapy in newly diagnosed glioblastoma failed to show any sign of activity, the further development of this anticancer drug was halted, and the interest in αVβ3 integrin inhibition as a therapeutic target has dwindled [6,7]. Likewise, the anti-αVβ3 antibody etaracizumab also failed to demonstrate significant therapeutic activity in patients with melanoma. In contrast, the anti-αV antibody abituzumab has shown modest activity in recurrent KRAS WT colorectal cancer, and is currently being tested in a larger randomized phase II clinical trial. Likewise, the β1 inhibitor volociximab, in spite of encouraging preliminary results, has also failed to demonstrate therapeutic benefits.

So, what went wrong in the development of integrins inhibitors as anti-cancer drugs? Is the choice of the target (i.e., the integrin) a problem? It is hard to imagine that the contribution of integrins to tumor development and progression observed in experimental and preclinical animal models does not apply to human cancer. Alternative, is it our still incomplete understanding of the complexity of integrin function and biology that that has mislead us toward overoptimistic approaches? Or, are the inhibitors used inappropriate? Could this be a “simple” pharmacokinetic issue, or a more “complex” problem associated with intrinsic properties of integrin that we failed to recognize? Maybe in the studied cancer types, integrins did not play a predominant role, or the lack of a biomarker for patient selection led to the failure. These are complex questions, for which today we do not have definitive answers. Nevertheless, they should stimulate us to think about developing new concepts, tools, and approaches to successfully exploit this fascinating class of molecules for the benefit of cancer patients (Figure 2).

### 7.1. Did We Target the Wrong Integrin(s)?

Based on experimental results, the large majority of the 24 known integrins is implicated in cancer progression [223,224]. However, so far, only a few integrins have been explored clinically as therapeutic targets in anti-angiogenic therapies: αVβ3 and αVβ5, with the small cyclic peptide cilengitide; the αV subfamily (i.e., αVβ1/αVβ3/αVβ5/αVβ6/αVβ8) with the pan anti-αV antibody abituzumab and intetumumab; and α5β1 with the anti-α5 antibody volociximab. The role of αVβ3 in tumor angiogenesis have been questioned by genetic evidence, demonstrating that the constitutive ablation of the β3 subunit increased tumor angiogenesis [149,150], while conditional deletion in growing tumors had no anti-tumor effect [151]. Besides, αVβ3 and αVβ5, which are endothelial cells, express additional integrins (i.e., α3β1, α4β1, α5β1, α6β1, α9β1, α6β4, α2β1) that were not systematically considered and tested as potential therapeutic targets. The functional redundancy, promiscuity, and compensation typical of integrins may be the reasons why these integrin inhibitors were well tolerated, but at the same time had limited therapeutic effects. Redundancy and functional compensation call for testing the concomitant inhibition of multiple integrins in preclinical models. However, multiple targeting, if effective in animal models may be difficult to achieve in patients.

### 7.2. Did We Use the Wrong Inhibitor(s)?

Traditionally, integrin inhibitors have been conceived and screened for their ability to interfere with cell adhesion and migration. These inhibitors generally target the extracellular domain to prevent ligand binding, either by competitive, high-affinity occupation of the ligand-binding pocket (e.g., cilengitide), or by preventing affinity maturation or sterically hindering ligand binding (e.g., antibodies) (Figure 3) [164,225]. However, integrins have complex features (i.e., allosteric regulation, including cis interaction with other cell surface receptors and clustering at focal complexes and interaction with cytoplasmic proteins), making them unique relative to other receptors that may complicate the generation of effective inhibitors [226]. For instance, high-affinity “inhibitors” can prevent ligand binding, but at the same time, can induce a “superactive” conformation sustaining cellular signaling and resulting in enhanced angiogenesis [152,153,227]. This may be particularly relevant to the clinical use of cilengitide, which has a short half-life of about two to four hours, and its concentration in the plasma fluctuates dramatically between injections, resulting in possible dual effects [7,28]. A further complication is suggested by the inability of Arg–Gly–Asp-based antagonists to disrupt αVβ3-ligand binding in contrast to allosteric antagonists [228]. Thus, allosteric antagonists may be more effective than competitive antagonists to interfere with ligand-occupied integrins. Also, it appears that ligand occupied integrins can still transduce survival signals, even in cells in suspension [59,60,61]. For example, treating cells with integrin ligand antagonists or expressing mutant integrins with impaired ligand-binding capacity did not prevent αVβ3 from promoting the CSC phenotype [95].

These observations raise the possibility that integrins disengaged from the ECM but occupied by natural ligands or high-affinity inhibitors may still signal and sustain cell survival. This hypothesis can be tested by either inhibiting residual integrin-dependent signaling (e.g., by using kinase inhibitors) in cells treated with integrin inhibitors, or by developing a novel class of integrin antagonists that block both cell adhesion and signaling. As the interaction between the cytoplasmic tails with cytoplasmic structural and signaling proteins is essential for both adhesion and signaling functions [229], this may open unexplored opportunities to develop a novel class of inhibitors. We previously showed that the expression of isolated β integrin subunit cytoplasmic and transmembrane domains in adherent endothelial cells in vitro and in vivo caused massive cell detachment and death. The mechanisms involved competition for the binding of essential cytoplasmic adaptor proteins (e.g., talin) to engaged integrins (dominant negative effect), resulting in a ‘mechanical uncoupling’ of the integrins from cytoskeletal structures and signaling molecules [230,231,232]. Unfortunately, the effect was not integrin-specific: the expression of isolated β3 or β1 subunit cytoplasmic domain indiscriminately blocked both β3 and β1 functions. This was likely because interactions of cytoplasmic domains with key proteins of the adhesome are largely conserved across different integrins. Consistent with these observations, point mutations in the Y747F and Y759F of β3 subunits inhibited tumor angiogenesis and tumor growth [233]. As the structure and molecular composition of the adhesion, as well as the molecular interactions controlling integrin function all have dramatically improved since [35,36], it may be worth revisiting the possibility of selectively inhibiting integrin function by interfering with the adhesome. For example, the deletion of kindlin-2 reduced endothelial sprouting, while ILK silencing reduced endothelial cell migration, tube formation, and tumor angiogenesis [234,235,236].

### 7.3. Did We Target the Wrong Biological Process(es)?

A key rationale that fostered the development of integrin inhibitors as anti-cancer agents, in particular those targeting αVβ3, was their ability to suppress tumor angiogenesis in preclinical models [237]. As we have learned from clinical trials with numerous anti-angiogenic drugs, the suppression of tumor angiogenesis alone was revealed to be insufficient to effectively control tumor progression. Eventually, only a few anti-angiogenic drugs were approved for clinical use, and this was largely in combination with cytotoxic chemotherapy, in a limited number of advanced and metastatic cancers. Moderate outcome benefits were demonstrated in colorectal, kidney, and liver cancer, while the benefits in lung, breast, and ovarian cancer or glioblastoma were very modest or absent [6,238]. Thus, it could be that the effects of integrins’ inhibition on the tumor vasculature did not translate into anti-tumor effects, not because of their inefficacy, but because of the complex relationship between tumor angiogenesis and tumor progression, and the ability of cancer cells to adapt and escape from anti-angiogenic therapies [163]. The combined targeting of several integrin receptors and other angiogenic pathways may be required to obtain significant anti-tumor effects.

Importantly, until today, we lacked an adequate biomarker to predict which patients are likely to benefit from anti-integrin treatment. While metabolic imaging and tissue analysis suggest that cilengitide reaches its target, there is no information about its effects on the tumor vasculature or tumor invasion in patients [7]. In the CENTRIC (Cilengitide, Temozolomide, and Radiation Therapy in Treating Patients With Newly Diagnosed Glioblastoma and Methylated Gene Promoter Status) trial (Methyl Guanine Methyl Transferase (MGMT) promoter methylated tumors), αVβ3 expression did not reveal any prognostic or predictive information, while in the CORE (Cilengitide, Temozolomide, and Radiation Therapy in Treating Patients With Newly Diagnosed Glioblastoma and Unmethylated Gene Promoter Status) trial (unmethylated tumors), higher tumor expression levels of αVβ3 were associated with slightly improved progression-free and overall survivals in patients treated with cilengitide [177]. Still, this puts doubt as to whether tissue integrin expression is a useful biomarker. The identification of biomarkers that are predictive of responses—or as a surrogate for target engagement and for monitoring the activity of integrin inhibitors—is needed before further clinical investigation is carried out.

The three main cancer cell-related processes that were considered as attractive targets for integrin inhibitors are cell proliferation, survival, and invasion [18,239]. However, impinging on cell proliferation and survival on cancer cells through integrin inhibition may be difficult to achieve, as fully transformed cells acquire cell-autonomous growth and survival capacities in the absence of extracellular cues through the activation of oncogenes and the inactivation of tumor suppressor genes [240]. The inhibition of invasion, which is a critical event in the metastatic cascade, seems more plausible to achieve. However, since patients treated with integrins inhibitors had advanced invasive or metastatic diseases, these potential benefits may be blunted. As metastasis represents a starting point for further seeding, one may nevertheless expect effects on further metastatic spreading [241,242]. However, it should also be noted that cancer cells can invade the ECM without the need for integrin engagement through ameboid movements, which obviously would represent an escape mechanism to integrin inhibition [35,56,57].

The microenvironment of most tumors is densely infiltrated with leucocytes and lymphocytes, which use integrins for their homing, migration, and local functions [243,244,245]. While targeting leucocyte integrins has been successful in the management of autoimmune diseases, it remains largely unexplored in cancer [35]. This is a potentially appealing strategy considering that the tumor microenvironment is rich in immunosuppressive cells (i.e., Regulatory T cells, Treg; Myeloid Derived Suppressor Cells, MDSC) and their inhibition boosts the immune response [246]. For example, antagonists of integrin α4β1 blocked the extravasation of monocytes into tumor tissue and prevented the monocyte macrophage colonization of tumors and tumor angiogenesis [247]. While attractive conceptually, a selective inhibition of Treg or MDSC recruitment may be difficult to achieve in practice. Immune and inflammatory cells in the tumor microenvironment (i.e., granulocytes, MDSC, monocytes, Dendritic Cells (DC) cells, T cell, B cells, Natural Killer (NK) cells) use overlapping integrins to home, extravasate, and migrate to the tumor microenvironment, namely αLβ2, αMβ2, αXβ2, α4β1, α1β1, α2β1, αVβ3, and α5β1. This overlapping pattern of expression and the targeting some of the integrins, most notably α4 and β2, may cause severe unwanted effects such as viral and bacterial infections, as these integrins are critical to the physiological immune response, which significantly complicates such an approach [35].

A further cell population of the tumor microenvironment that may be considered to be targeted with integrins inhibitors is the CAF. CAF are well established to promote cancer progression through paracrine communication and remodeling of the ECM, which is a process that is tightly dependent on integrins. Thus, inhibiting integrins on CAF, in particular of the αV and β1 family, is likely to interfere with matrix deposition and remodeling, and indirectly to suppress matrix-dependent functions that normally promote tumor progression, in particular survival and invasion [31,32].

In addition to inhibiting integrin function to impinge on cancer-associated events, integrins may serve as targets to deliver cytotoxic drugs to tumors, for instance by using antibody–drug conjugates (ADCs). While this approach is not fully novel, it may pay off in some cancers, and may be combined with diagnostic procedures (theragnostic).

### 7.4. Did We Use the Wrong Preclinical Models?

Preclinical tumor models used for drug testing are typically based on monitoring drug activity on primary tumor growth and possibly on their metastatic dissemination [248,249]. These conditions do not reflect the clinical situation in which systemic therapy is either used as adjuvant therapy to eliminate microscopic or minimal residual disease and/or already disseminated cancer cells, or as palliative therapies in patients with advanced and metastatic disease. Thus, most preclinical models do not represent the actual clinical situation of patients eventually receiving the tested drug at the advanced/metastatic stages [249]. This is a potentially confounding factor that may lead to overestimating the therapeutic effects of the tested drugs. In addition, transplantable tumors based on cell lines do not fully recapitulate the biological and molecular feature of the corresponding human tumors, and in the case of xenografts, the host is immunodeficient, thereby depriving the tumor microenvironment of important immunological functions. As this is a general problem in drug testing, there is an urgent need to align preclinical models with the relevant clinical situation. Suitable models should include orthotopic primary tumor growth, spontaneous metastasis formation, primary tumor removal, and the treatment of metastatic disease.

### 7.5. Did We Perform the Wrong Clinical Trials?

A number of clinical trials were performed for over a decade, investigating either competitive peptide-based inhibitors or monoclonal antibodies targeting specific integrins (Table 2). The largest trial portfolio was for cilengitide, which is a cyclic inhibitor targeting αVβ3 and αVβ5 integrins. Trials were well planned, and many were designed as randomized phase 2 trials aiming at identifying the optimal dose or dosing regimen and activity signal seeking. Nevertheless, with a short half-life of only a few hours, a twice-weekly infusion schedule may not have been an optimal choice, while drug solubility limited the options of a continuous infusion schedule. The only partial penetration through the blood–brain barrier is another limitation in particular when treating primary brain tumors. The cilengitide development program also illustrates the limitations of randomized phase II trials that cannot replace formal comparative phase III trials. Furthermore, a more critical interpretation of phase II results and the integration of correlative endpoints is warranted if this design is to determine the fate of an investigational agent. Predefined stringent criteria for “success” and “go/no go” decisions should be determined at the onset of phase II trials.

Most trials targeting integrins to date are similar in design, and have repeatedly focused on the same clinically defined tumor entities. Although some of the trials are aimed at enriching a certain molecular subgroup (e.g., MGMT methylation in glioblastoma trials, mutated Kirsten RAS (KRAS) in colorectal cancer), these markers are of important prognostic value of the disease of interest. However, they have no mechanistic relationship to integrins as a treatment target.

## 8. Conclusions

In spite of compelling experimental results demonstrating that integrins contribute to cancer progression and that their inhibition has therapeutic effects, clinical trials with αVβ3/αVβ5 and α5β1 integrin inhibitors have globally failed to demonstrate therapeutic benefits, and no inhibitors has been registered as anti-cancer drug. However, therapy strategies have focused so far solely on integrins on tumor cells and vascular cells. Consequently, new approaches targeting integrins in other cells of the tumor microenvironment, e.g., cancer-associated fibroblasts and inflammatory/immune cells, are necessary and should be considered. Additionally, the pharmacological properties of the integrin inhibitor and the heterogeneity and redundancies of integrin functions require further understanding before proceeding with future investigation of novel integrin-targeting agents in the clinic. We conclude that integrins remain a valid target for cancer therapy, but novel preclinical models and translational studies focusing on the tumor microenvironment are needed.

## Figures and Tables

**Figure 1 cancers-11-00978-f001:**
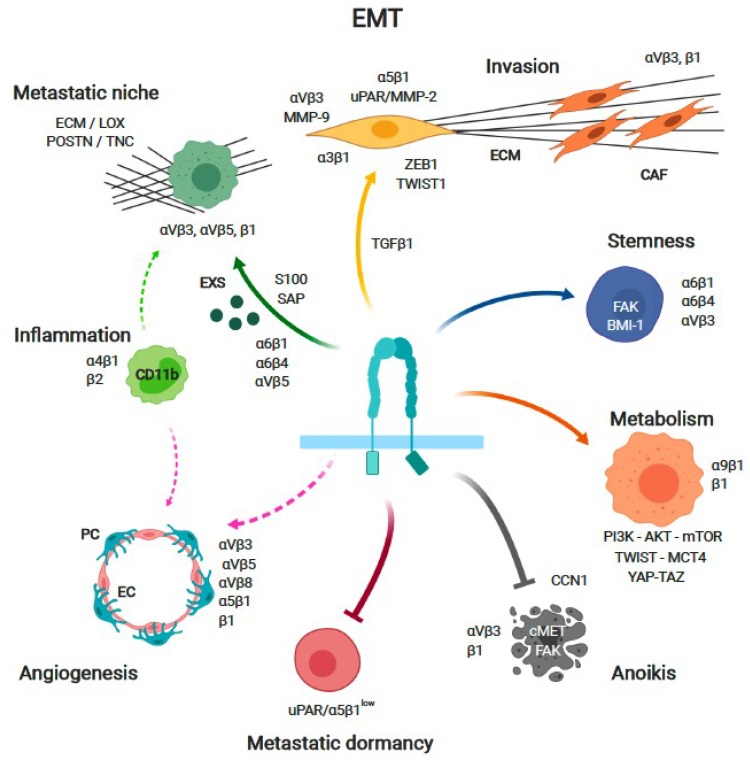
Integrin-dependent functions relevant to cancer. Integrins have been implicated in mediating several hallmarks of cancer, including cancer cell proliferation, dormancy, survival, stemness, metabolic adaptation, and metastatic niche formation. Integrins also promote epithelial-to-mesenchymal transition and invasion, which are two key steps of metastasis formation. In the tumor microenvironment, integrins promote endothelial cell survival and angiogenesis, the recruitment of immune and inflammatory cells, and stroma remodeling and fibrosis induced by cancer-associated fibroblasts. The role of integrins in these functions are described in more detail in Section 4.1–Section 4.6 and Section 5.1, and Section 5.2. This listing is non-exhaustive. Abbreviations: CAF, cancer-associated fibroblasts; CCN1, cysteine rich protein 61 (CYR61); CSC, cancer stem cell; endothelial cell; EC, endothelial cells, ECM, extracellular matrix; EMT, epithelial to mesenchymal transition; EXS, exosomes; LOX, lysyl oxidase; PC, pericytes; POSTN, periostin; TC, tumor cell; TNC, tenascin.

**Figure 2 cancers-11-00978-f002:**
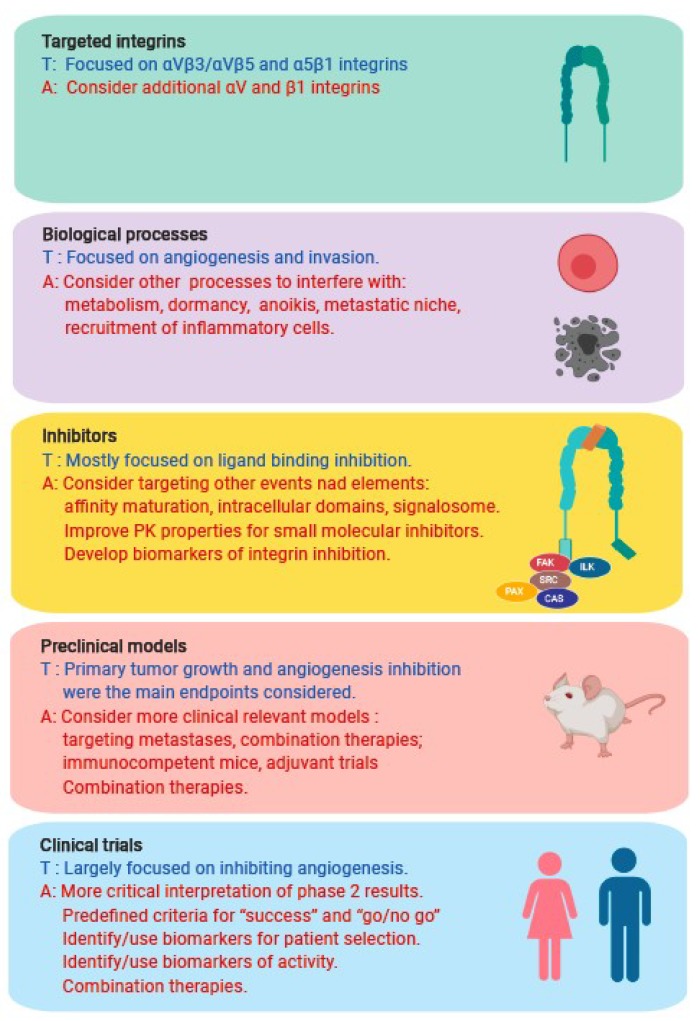
What went wrong with the development of integrin inhibitors in cancer, and what can we do different? The field focused largely on a few integrins, most notably αVβ3/αVβ5 and α5β1 based on early preclinical work with the purpose to target tumor angiogenesis, using a limited set of inhibitors (mostly interfering with ligand binding). A better understanding of integrin function and biology, and the accumulated experience with clinical studies, should stimulate us to think about developing new concepts, tools, and approaches to successfully exploit integrins as therapeutic targets in cancer. Here is a non-exhaustive summary of the concepts discussed in the text. T, tested in the past to present; A, alternative strategies to consider.

**Figure 3 cancers-11-00978-f003:**
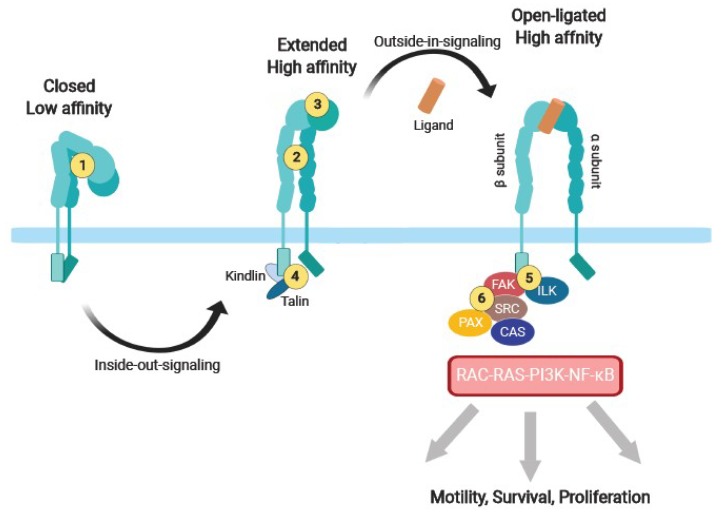
Alternative strategies to inhibit integrins. Current integrin inhibitors are mostly based on preventing ligand binding through direct competition or (allo)steric interference with the extracellular domains. However, some of these ligand-binding inhibitors may activate integrins and do not fully suppress integrin signaling. Alternative approaches to interfere with integrin function may be considered since integrin activation and signaling are complex events involving multiple and different steps. Strategies to consider include: 1, the retention of bent integrins in their low-affinity state; 2, the prevention of full integrin extension and affinity maturation; and 3, irreversible preclusion of ligand binding by covalent modification of the binding pocket. The intracellular domains and the adhesome also provide additional opportunities, including: 4, interfering with kindlin/talin-mediated activation; 5, the prevention of recruitments of signaling proteins of the adhesome (e.g., FAK) to the β cytodomain; 6, the prevention of adhesome maturation or induction of adhesome dissolution by interfering with protein–protein interactions.

**Table 1 cancers-11-00978-t001:** Overview of integrins expressed in cancer cells and the cells of the tumor microenvironment. The table lists the main integrins reported to play a role in cancer. For further reading, we refer to specific reviews and original articles [9,12,13,15,16,17,18,19,20,21,22,23,24,25,26,27,28,29,30,31,32]. Abbreviations: CAF, Cancer Associated Fibroblasts, MyF, Myofibroblasts.

Integrin Heterodimer	Arg-Gly-Asp Ligand Binding Dependency	Integrin Expression Patterns			
		Cancer Cells	Vascular Cells	CAF, MyF	Immune Cells
α1β1		+	++	++	++
α2β1		+++	++	++	++
α3β1		+++	++	++	
α4β1		+++		++	+++
α5β1	+	+++	+++	++	++
α6β1		+++	++		++
α7β1		++			
α8β1	+	+	++		
α9β1		++	++	++	
α10β1		++			
α11β1				++	
αVβ1	+	++	++	++	
αLβ2					+++
αMβ2					+++
αXβ2					+++
αDβ2					+++
αVβ3	+	+++	+++	++	+++
αiibβ3	+				Platelets
α6β4		+++	++		
αVβ5	+	+++	+++	++	
αVβ6	+	+++		++	
α4β7				+	+++
αEβ7					+++
αVβ8	+	++	+++	++	++

**Table 2 cancers-11-00978-t002:** Selected clinical trials of agents targeting integrins. Non-exhaustive listing of the recent most important clinical studies with integrin inhibitors and their salient features and results).

Study Name and Description	Indication	Phase/N pts	Design	Endpoints	Outcome and Remarks	References
**Abituzumab (EMD 525797, anti-αV integrin antibody) (Merck-Serono): total no. of trials 3**						
POSSEIDON: SofC ± abituzumab (two doses)	Colon Ca (KRAS WT)	II 216	dose finding/randomized	1^o^: PFS 2^o^: OS	No diff in PFS, superior surv. of both abituzumab arms vs. SoC.	[213]
AMELION: Cetuximab/FOLFIRI ± Abituzumab, high αVβ6 expr.	Colon Ca	II 230	Randomized	1^o^: PFS 2^o^: OS, RR	Start planned for 2nd quarter 2019	NA
**Intetumumab (CNTO95, anti-αV integrin antibody) (Centocor, Johnson &Johnson): total no. of trials: 3**						
Intetuzumab ± DTIC vs. DTIC	Melanoma	II 129	randomized (4-arms)	1^o^: PFS: 2^o^: OS, RR	Trend for improved OS with high-dose intetumumab	[214]
Docetaxel ± intetumumab	Prostate Ca	II 131	Randomized	1^o^: PFS 2^o^: RR	Outcome favors placebo (!)	[215]
**Cilengitide (EMD 121974, anti-αVβ3/αVβ5 integrin cyclic peptide) (Merck-Serono): total no. of trials: 21 (+ 8 terminated)**						
ADVANTAGE: CDDP/5-FU/Cetuximab ± cil weekly vs. 2×/wk vs. control	Rec/metast. H&NCa	II 184	3-arms	1^o^: PFS: 2^o^: OS, RR	No difference in 1^o^ or 2^o^ endpoints	[216]
CERTO: CDDP-based regimen ± cilengitide weekly or 2×/week	NSCLC	II 169	Randomized/dose-finding	1^o^: PFS 2^o^: OS	Inconsistent results	[217]
NABTT:0306: Cil 500 vs. 2000 mg + TMZ/RT→TMZ	nd GBM	II 112	Randomized	OS	Both arms improved over historical controls	[218]
Cil 500 vs. 2000 mg	Rec GBM	II 81	Randomized	PFS_6mo_	Responses at all doses	[219]
Cil 2000 mg	Prostate	II 16	Uncontrolled, 2-stage design	PSA response	No activity	[220]
010: Cil (500 mg) + TMZ/RT →TMZ	nd GBM	II 52	Pilot study, uncontrolled	1^o^: PFS_6mo_ 2^o^: OS	Comparison to historical control	[221]
CENTRIC: TMZ/RT→ TMZ ± Cil	Methyl. MGMT GBM	III 545	Pivotal international EORTC trial.	OS	No activity	[6]
CORE: Cil 5d/week vs. 2d/wk vs. control + TMZ/RT	Unmethyl. MGMT GBM	II 265	3-arms	OS 2: PFS	No differences	[222]
**Etaracizumab (MEDI-522, anti-αVβ3 integrin antibody) (MedImmune, Astra Zeneca): total trial 9 (+1 discontinued early)**						
Etatacizumab ± DTIC	Melanoma	II 112	Randomized	RR, OS	No responses with etatcizumab alone. No further evaluation recommended.	[165]
**Volociximab (MEDI-522, anti α5β1 integrin antibody) (AbbVie): total trials 7 (+ 3 discontinued early)**						
Numerous uncontrolled phase II studies against lung, pancreatic and ovarian cancer						
SoC; standard of care. FOLFIRI; 5FU, leucovorin, irinotecan. DTCI; dacarbazine. CDDP; cisplatin. TMZ; temozolomide. RT; radiotherapy. Cil; cilengitide. PFS; progression-free survival. OS; overall survival. RR; response rate. 1^o^; primary endpoint. 2^o^, secondary endpoint. Rec, recurrent; nd, newly diagnosed						

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
