# Peer review of "Are Integrins Still Practicable Targets for Anti-Cancer Therapy?"

_cancers, 2019, doi:10.3390/cancers11070978_

Reviewer 1 Report

The authors provide a critical and detailed overview of integrin functions and potential diagnostic and therapeutic applications.   

Unfortunately, the interested reader will not be able to understand figure 1 without a profound background knowledge or without a detailed description of the symbols. The authors can not expect the readers to comprehend, how figure 1 was intended to provide supporting information to the text.

On the other hand, the text does not give enough information with regard to figure 1 that the reader perceives figure 1 as a helpful addition. Beside, the quality of the drawings and the organization of the symbols does not match the high quality of the journal.

Regarding figure 2:

The depicted illustration for “targeted integrins” are too small. The quality of the drawings must be improved or provided by skilled and/or trained staff for figure illustration. The present design of the figure does not meet the high standards of the journal.

Please reconsider the sentence, if the authors intended the term "all-anti anti-angiogenic therapies".

The reviewer would suggest a simpler sentence structure.

"It should also be mentioned that the failure of cilengitide [134] to block tumor angiogenesis and tumor progression may not be specific to integrin inhibitors, but may involve mechanisms of resistance common to all-anti anti-angiogenic therapies [152].

Please revise citation in the following sentence:

αVβ3, emerged as useful 375 targets to this end Arg-Gly-Asp-based cyclic peptides targeting the tumor vasculature or tumor cells 376 have been successfully used to deliver conjugated nanocarriers or liposomes loaded with 377 chemotherapeutic drugs, that may be combined also for imaging purposes (theranostics) Danhier, 378 2012 #293;Gajbhiye, 2018 #283}.

Please revise citation in the following sentence:

Or is the complexity of integrin function and biology that that are misleading 442 our simple minded “one integrin – one function – one effect” linear approach?

 For the chapter 6.3 Tumor imaging please consider to refer to the work:

Fluorescence imaging of invasive head and neck carcinoma cells with integrin αvβ6-targeting RGD-peptides: an approach to a fluorescence-assisted intraoperative cytological assessment of bony resection margins.

Nieberler M, Reuning U, Kessler H, Reichart F, Weirich G, Wolff KD.

Br J Oral Maxillofac Surg. 2018 Dec;56(10):972-978. doi: 10.1016/j.bjoms.2018.11.003. Epub 2018 Nov 28.

PMID:30502043

The above mentioned application is based on the avß6-specific RGD ligand, as described in:

Stable Peptides Instead of Stapled Peptides: Highly Potent αvβ6-Selective Integrin Ligands.

Maltsev OV, Marelli UK, Kapp TG, Di Leva FS, Di Maro S, Nieberler M, Reuning U, Schwaiger M, Novellino E, Marinelli L, Kessler H.

Angew Chem Int Ed Engl. 2016 Jan 22;55(4):1535-9. doi: 10.1002/anie.201508709. Epub 2015 Dec 9.

PMID:26663660

Overall the reviewer suggests that the manuscript should be spell checked and the above-mentioned points should be taken in to account, before the manuscript is further considered for publication.

Author Response

REVIEWER 1

The authors provide a critical and detailed overview of integrin functions and potential diagnostic and therapeutic applications.   

Unfortunately, the interested reader will not be able to understand figure 1 without a profound background knowledge or without a detailed description of the symbols. The authors can not expect the readers to comprehend, how figure 1 was intended to provide supporting information to the text. On the other hand, the text does not give enough information with regard to figure 1 that the reader perceives figure 1 as a helpful addition. Beside, the quality of the drawings and the organization of the symbols does not match the high quality of the journal.

It is indeed a complex figure. We have modified the figure legends as following to make it more explicative:

Figure 1. Integrin-dependent functions relevant to cancer. Integrins have been implicated in mediating several hallmarks of cancer, including cancer cell proliferation, dormancy, survival, stemness, metabolic adaptation, and metastatic niche formation. Integrins also promote epithelial-to-mesenchymal transition and invasion, two key steps of metastasis formation. In the tumor microenvironment, integrins promote endothelial cell survival and angiogenesis, recruitment of immune and inflammatory cells, and stroma remodeling and fibrosis induced by cancer associated fibroblasts. The role of integrins in these functions are described more in details in sections 4.1-4.6 and 5.1, 5.2. The listing is non-exhaustive. Abbreviations: CAF, cancer associated fibroblasts; CCN1, cysteine rich protein 61 (CYR61); EC, CSC, cancer stem cell; endothelial cell; ECM, extracellular matrix; EMT, epithelial to mesenchymal transition; EXS, exosomes; LOX, lysyl oxidase; PC, pericyte; POSTN, periostin; TC, tumor cell; TNC, tenascin.

We also modified the figure and used a professional graphic program. We put an integrin at the center of the figure to depict that the mentioned functions are integrin-mediated functions. 

We have cited the figures multiple times in the text, at relevant paragraphs. We hope this figure is now better understandable.

Regarding figure 2:

The depicted illustration for “targeted integrins” are too small. The quality of the drawings must be improved or provided by skilled and/or trained staff for figure illustration. The present design of the figure does not meet the high standards of the journal.

We made a new figure by using a professional graphic program. We are made a new figure 3.

Please reconsider the sentence, if the authors intended the term "all-anti anti-angiogenic therapies".

The reviewer would suggest a simpler sentence structure. "It should also be mentioned that the failure of cilengitide [134] to block tumor angiogenesis and tumor progression may not be specific to integrin inhibitors, but may involve mechanisms of resistance common to all-anti anti-angiogenic therapies [152].

We have changed the sentence as follows::

Considering the failures of other anti-angiogenic therapies (e.g. the anti-VEGF antibody bevacizumab) in halting glioblastoma progression, the failure of cilengitide as anti-angiogenic drug [144] may not be due to cilengitide itself, but to the problem of evasive resistance intrinsic to all anti-angiogenic therapies [162-164].

Please revise citation in the following sentence:

αVβ3, emerged as useful 375 targets to this end Arg-Gly-Asp-based cyclic peptides targeting the tumor vasculature or tumor cells 376 have been successfully used to deliver conjugated nanocarriers or liposomes loaded with 377 chemotherapeutic drugs, that may be combined also for imaging purposes (theranostics) {Danhier, 378 2012 #293;Gajbhiye, 2018 #283}.

This was a formatting issue which we overlooked in the prior submission, thank you for highlighting. We revised and updated the sentence as follows:

 “For example, αVβ3-specific Arg-Gly-Asp-based cyclic peptides targeting the tumor vasculature or tumor cells have been successfully used to deliver therapeutic compounds, but also to image tumor lesions (theranostics) [184,185].

Please revise citation in the following sentence:

Or is the complexity of integrin function and biology that that are misleading 442 our simple minded “one integrin – one function – one effect” linear approach?

We have modified the sentence as following.

“Or is our still incomplete understanding of the complexity of integrin function and biology that that has mislead us toward overoptimistic therapeutic approaches?”

 For the chapter 6.3 Tumor imaging please consider to refer to the work:

Fluorescence imaging of invasive head and neck carcinoma cells with integrin αvβ6-targeting RGD-peptides: an approach to a fluorescence-assisted intraoperative cytological assessment of bony resection margins.

Nieberler M, Reuning U, Kessler H, Reichart F, Weirich G, Wolff KD.Br J Oral Maxillofac Surg. 2018 Dec;56(10):972-978. doi: 10.1016/j.bjoms.2018.11.003. Epub 2018 Nov 28. PMID:30502043

The above mentioned application is based on the avß6-specific RGD ligand, as described in:

Stable Peptides Instead of Stapled Peptides: Highly Potent αvβ6-Selective Integrin Ligands.

Maltsev OV, Marelli UK, Kapp TG, Di Leva FS, Di Maro S, Nieberler M, Reuning U, Schwaiger M, Novellino E, Marinelli L, Kessler H.Angew Chem Int Ed Engl. 2016 Jan 22;55(4):1535-9. doi: 10.1002/anie.201508709. Epub 2015 Dec 9. PMID:26663660

We thank the reviewer for pointing to the use of this avß6-specific RGD ligand for imaging purposes. We have modified the paragraph and added these references in chapter 6.3

Overall the reviewer suggests that the manuscript should be spell checked and the above-mentioned points should be taken in to account, before the manuscript is further considered for publication.

We have carefully checked the spelling of the manuscript again.

Reviewer 2 Report

This manuscript is a well organized and complete review on the involvement of integrins in all the stages of cancer progression and represents a very useful overview both for scientist working in this filed and for researchers that want to face this topic for the first time. The English language is really fluent and easy to read. Some minor modifications are suggested:

Figure 1 is not cited in the main text. I suggest to introduce citation at line 76. 

Citation of figure 2 at line 496 is wrong, since figure 3 is described in this paragraph

After the introduction, a short paragraph presenting the different families of integrins, divided on the basis of the overexpressing cells (cancer cells, leukocytes, platelets, etc) could be useful to better understand which kind of scenarios is described in the following paragraphs

In section 4 and 5, different integrins' involving processes are described. Even if, at the end of each sub-section, the authors gave some hint to the possible new perspective, a new paragraph summarizing those areas that still deserve attention could be added.

In section 6, a clear distinction between monoclonal antibodies and small molecules should be introduced.

Line 371, reference 153 should be cited together with 9

Figure 2, a column indicating the targeted receptor for each study should be added

Line 487, "reasons" is wrongly spelled

On the basis of these comments, I suggest publication after minor revisions.

Author Response

REVIEWER 2

This manuscript is a well-organized and complete review on the involvement of integrins in all the stages of cancer progression and represents a very useful overview both for scientist working in this filed and for researchers that want to face this topic for the first time. The English language is really fluent and easy to read. Some minor modifications are suggested:

We thank this reviewer for her/his positive comments and suggestions.

Figure 1 is not cited in the main text. I suggest to introduce citation at line 76. 

Thank you for spotting this mistake. We have cited Figure 1 in the text at page 3, line 101.

Citation of figure 2 at line 496 is wrong, since figure 3 is described in this paragraph.

Yes indeed, we have corrected “figure 2” to “figure 3”

After the introduction, a short paragraph presenting the different families of integrins, divided on the basis of the overexpressing cells (cancer cells, leukocytes, platelets, etc) could be useful to better understand which kind of scenarios is described in the following paragraphs

We have created a new Table 1, listing all integrins involved in cancer progression, introduced at the end of section 2.

In section 4 and 5, different integrins' involving processes are described. Even if, at the end of each sub-section, the authors gave some hint to the possible new perspective, a new paragraph summarizing those areas that still deserve attention could be added.

We have added this paragraph at the end of section 5:

In summary, integrin inhibition as anti-cancer therapy was initially conceived based on their ability to promote cancer cell invasion, metastasis and tumor angiogenesis. Recent developments indicate that additional functions relevant to cancer cells are mediated or regulated by integrins, including dormancy, metabolism, survival, therapy resistance, EMT, fibrosis, cancer cell stemness, exosome homing and pre-metastatic niche formation. Thus, in the future it will be important to further understand the contribution of integrins to these emerging functions, and to evaluate the potential therapeutic impact of impinging  on these functions via integrin inhibition. “

In section 6, a clear distinction between monoclonal antibodies and small molecules should be introduced.

We have clarified in all cases whether antibodies, peptides or peptidomimetics were used. We have also further specified that nature of antagonists used as following:

“Antibodies, endogenous proteins or peptidic antagonists, synthetic peptides and peptidomimetics have been used to target integrins in cancer”. (lines 354-355)

Line 371, reference 153 should be cited together with 9

We have added Ref 153 to Ref 3 (updated as [9,165])

Also, at line 397 we have formatted two references that were not properly formatted.

Figure 2, a column indicating the targeted receptor for each study should be added

We assume the reviewer meant Table 1. The target was indicated already but may be not clearly visible. We have now indicated it next to the name of the drug, like this.

 “Abituzumab (EMD 525797, anti-αV integrin antibody) (Merck-Serono): total no of trials 3”

Line 487, "reasons" is wrongly spelled

Thank you for spotting we have corrected it.

On the basis of these comments, I suggest publication after minor revisions.

Thank you!

Reviewer 3 Report

The authors summarize information on preclinical and clinical results using certain integrins as therapeutic targets in cancer. They conclude that in spite of compelling preclinical results, inhibitors have failed in clinical trials and thus integrins as therapeutic targets and how to interfere with their function should be reconsidered. The review discusses what went wrong and what could be done differently. 

The authors should make clear, also in the abstract (and maybe in title) that this review focuses mainly on results from targeting the integrins avb3 and a5b1. The way it is written, about integrins in general, can be misleading.  

Alternatively, include information about other integrin targets and more recent publications. 

Results from using Antibody-Drug-Conjugates (ADCs) to target integrins could be included to strengthen the discussion on integrin drugs in cancer therapy. 

It would help to write out in Table 1 what integrin that is targeted in each case.

Paragraphs 6.2 and 6.3: Take out or shorten. Relates more to methods.

Author Response

REVIEWER 3

The authors summarize information on preclinical and clinical results using certain integrins as therapeutic targets in cancer. They conclude that in spite of compelling preclinical results, inhibitors have failed in clinical trials and thus integrins as therapeutic targets and how to interfere with their function should be reconsidered. The review discusses what went wrong and what could be done differently. 

The authors should make clear, also in the abstract (and maybe in title) that this review focuses mainly on results from targeting the integrins avb3 and a5b1. The way it is written, about integrins in general, can be misleading.  Alternatively, include information about other integrin targets and more recent publications. 

Indeed, results of therapeutic integrin targeting was mostly based on avb3/avb5 and a5b1. We have therefore highlighted this fact in the summary and in the introduction

Results from using Antibody-Drug-Conjugates (ADCs) to target integrins could be included to strengthen the discussion on integrin drugs in cancer therapy. 

Some of these results are briefly included in paragraph 6.2, however we feel that further details are beyond the scope of this already complex and extensive review. We have nevertheless added a sentence in the discussion sections at point 7.3:

In addition to inhibiting integrin function to impinge on cancer-associated events, integrins may serve as targets to deliver cytotoxic drugs to tumors, for instance by using antibody-drug-conjugates (ADCs). While this approach is not fully novel, it may pay off in some cancers and may be combined with diagnostic procedures (theragnostic). “

It would help to write out in Table 1 what integrin that is targeted in each case.

The target was indicated already but may be not clearly visible. We have now indicated it next to the name of the drug, like this.

 “Abituzumab (EMD 525797, anti-αV integrin antibody) (Merck-Serono): total no of trials 3”

Paragraphs 6.2 and 6.3: Take out or shorten. Relates more to methods.

We have indeed shortened these paragraphs by removing technical details.

Round  2

Reviewer 1 Report

Dear authors, the reviewer appreciates the quick corrections and the improvement to the figures.

The manuscript in the current form is more appealing to the reader as the previous version.

The critical points have been addressed in an adequate manner.